# Improving sleep and learning in rehabilitation after stroke, part 2 (INSPIRES2): study protocol for a home-based randomised control trial of digital cognitive behavioural therapy (dCBT) for insomnia

Matthew Weightman ![ORCID],[1] Barbara Robinson ![ORCID],[1] Ricky Fallows,[2] Alasdair L Henry,[3,4] Simon D Kyle ![ORCID],[4] Emma Garratt,[5] Anton Pick,[6] Rachel Teal,[7] Sara Ajina,[8,9] Nele Demeyere,[10,11] Colin A Espie,[3,4] Ben Seymour,[1] Heidi Johansen-Berg,[1] Melanie K Fleming ![ORCID][1]

For numbered affiliations see end of article.

**Correspondence to**
Dr Matthew Weightman;
matthew.weightman@ndcn.ox.ac.uk

## ABSTRACT

**Introduction** Consolidation of motor skill learning, a key component of rehabilitation post-stroke, is known to be sleep dependent. However, disrupted sleep is highly prevalent after stroke and is often associated with poor motor recovery and quality of life. Previous research has shown that digital cognitive behavioural therapy (dCBT) for insomnia can be effective at improving sleep quality after stroke. Therefore, the aim of this trial is to evaluate the potential for sleep improvement using a dCBT programme, to improve rehabilitation outcomes after stroke.

**Methods and analysis** We will conduct a parallel-arm randomised controlled trial of dCBT (Sleepio) versus treatment as usual among individuals following stroke affecting the upper limb. Up to 100 participants will be randomly allocated (2:1) into either the intervention (6–8 week dCBT) or control (continued treatment as usual) group. The primary outcome of the study will be change in insomnia symptoms pre to post intervention compared with treatment as usual. Secondary outcomes include improvement in overnight motor memory consolidation and sleep measures between intervention groups, correlations between changes in sleep behaviour and overnight motor memory consolidation in the dCBT group and changes in symptoms of depression and fatigue between the dCBT and control groups. Analysis of covariance models and correlations will be used to analyse data from the primary and secondary outcomes.

**Ethics and dissemination** The study has received approval from the National Research Ethics Service (22/EM/0080), Health Research Authority (HRA) and Health and Care Research Wales (HCRW), IRAS ID: 306 291. The results of this trial will be disseminated via presentations at scientific conferences, peer-reviewed publication, public engagement events, stakeholder organisations and other forms of media where appropriate.

**Trial registration number** NCT05511285.

### STRENGTHS AND LIMITATIONS OF THIS STUDY

⇒ Use of an evidence-based therapy to improve sleep symptoms.
⇒ Home-based intervention, which reduces burden on participants.
⇒ Unable to blind participants to the treatment group.
⇒ Motor training component of the trial is optional and, therefore, will likely have a reduced sample size.

## INTRODUCTION

Stroke is one of the leading causes of long-term disability worldwide, often resulting in prolonged cognitive, motor and emotional problems.[1] Many stroke survivors also experience long-term sleep disruption,[2–4] with the prevalence of post-stroke insomnia, a sleep disorder characterised by difficulties initiating and maintaining sleep, estimated to be as high as 60%.[5–7] Poor sleep after stroke is frequently associated with motor/cognitive impairments, mood disorders such as anxiety and depression and a reduced quality of life.[8–10] There is also appreciable evidence to suggest that those who have poorer sleep show worse rehabilitation outcomes,[8 11] which we hypothesise may be, at least in part, due to the importance of sleep for learning and memory processes.

Motor learning—the process of learning new, or relearning old, motor skills—is an integral part of post-stroke rehabilitation.[12] Improvements in motor performance not only occur during practise itself but also during periods of rest, including sleep.[13] In recent years, a large body of evidence

BMJ

has emerged, indicating these latter improvements in performance, better known as consolidation, are sleep-dependent.[14–21] Mechanistically, the consolidation of motor learning during sleep is thought to occur via 'reactivations' of neural activity elicited during initial learning, with stronger reactivation correlating to a greater degree of memory consolidation.[20 22] Therefore, it stands to reason that consolidation of motor memories after or between therapy sessions may be impaired post-stroke, via direct disruption of these biological processes that occur during sleep, thus hindering recovery. Indeed, recent results from our lab have demonstrated that sleep disturbances during rehabilitation after acquired brain injury are associated with poorer motor recovery outcomes.[8] Together these findings suggest that improving sleep quality post-stroke could potentially also help to enhance rehabilitation outcomes. However, research and interventions focused on improving symptoms of post-stroke insomnia are lacking.

Cognitive behavioural therapy (CBT) is a National Institute for Health and Care Excellence (NICE) recommended, first-line treatment for those with insomnia disorder[23–25] and is supported by a wealth of evidence for improving symptoms in otherwise healthy individuals.[26 27] Considering that few patients receive access to therapist-delivered CBT due to systemic barriers,[28 29] fully-automated digital CBT (dCBT) has the potential to provide access to evidence-based treatment at scale. We, and others, have shown that the delivery of dCBT can improve insomnia symptoms,[30 31] psychological well-being,[32 33] cognitive function[34] and is also feasible to use after stroke.[35] Furthermore, preliminary data suggest that dCBT is also efficacious at improving self-reported measures of sleep and mood in adult, community dwelling, stroke survivors when compared with provision of sleep hygiene information.[36]

## Study objectives

Given the possible benefit of improved sleep on recovery of motoric function after stroke, the overall aim of the present study is to explore the potential for sleep improvement, via dCBT, to impact on rehabilitation outcomes post-stroke. Termed INSPIRES-2 (improving sleep and learning in rehabilitation after stroke, part 2), we will more specifically aim to:

1. Test whether symptoms of insomnia are improved in stroke survivors following dCBT in comparison to treatment as usual.
2. Assess whether overnight consolidation of motor learning is improved after the delivery of dCBT in comparison to treatment as usual.
3. Determine whether actigraphy-derived sleep measures are improved after the delivery of dCBT in comparison to treatment as usual.
4. Identify whether changes in sleep following dCBT correlate with changes in overnight consolidation of motor learning.

5. Test for changes in symptoms of depression and fatigue after dCBT in comparison with treatment as usual.
6. Explore the feasibility and potential effect size estimates of at-home motor training following dCBT or usual care.

## METHODS AND ANALYSIS

### Participant recruitment

Up to 100 participants with stroke affecting the upper limb will be recruited. Participants will be identified via: National Health Service (NHS) patient identification centres, from prior study involvement at the Wellcome Centre for Integrative Neuroimaging (WIN) or Cognitive Neuropsychology Centre (University of Oxford), social media, stroke support groups and word of mouth. Participants may be identified at any time following their stroke but will not be enrolled in the study until they have been discharged from inpatient care. There is no specific post-stroke period restriction for recruitment to this trial.

The trial opened to recruitment in September 2022, with a planned trial end date of December 2024. The study was registered on 22 August 2022 (prior to recruitment of the first participant).

### Inclusion/exclusion criteria

Participants must be 18 years of age or above, discharged from inpatient care and have a clinical diagnosis of stroke affecting the upper limb. Participants should, however, have sufficient movement to be able to perform the prescribed motor learning task. Individuals must be willing and able to give informed consent for participation in the study. Participants should also be interested in accessing a programme intentionally aimed at improving sleep quality and given the nature of the study, have reliable access to the internet. Individuals may not enter the study if any of the following apply: diagnosis of other neurological condition/s affecting movement (eg, Parkinson's disease) or untreated diagnosed sleep disorder/s (eg, sleep apnoea); experience uncontrolled seizures; have engaged in psychological therapy for insomnia in the past 12 months; are pregnant or have planned inpatient admission (eg, for rehabilitation) in the next 4 months that would impact on the ability to engage with the dCBT programme.

### Sample size distribution

Based on prior work in our lab comparing dCBT with sleep hygiene information in stroke survivors[36] and other studies involving dCBT in people who suffer from insomnia, we estimate a medium between group effect size of d=0.7. In order to achieve this, a sample size of n=76 will be required ($\alpha$=0.05, power=80%, 2:1 random allocation ratio dCBT:control, two-tailed independent samples t-test). A sample size of n=100 was selected to accommodate an approximate 25% dropout rate (determined from previous research experience). As the at-home motor training component of the trial is optional, we expect the resultant sample size to be reduced.

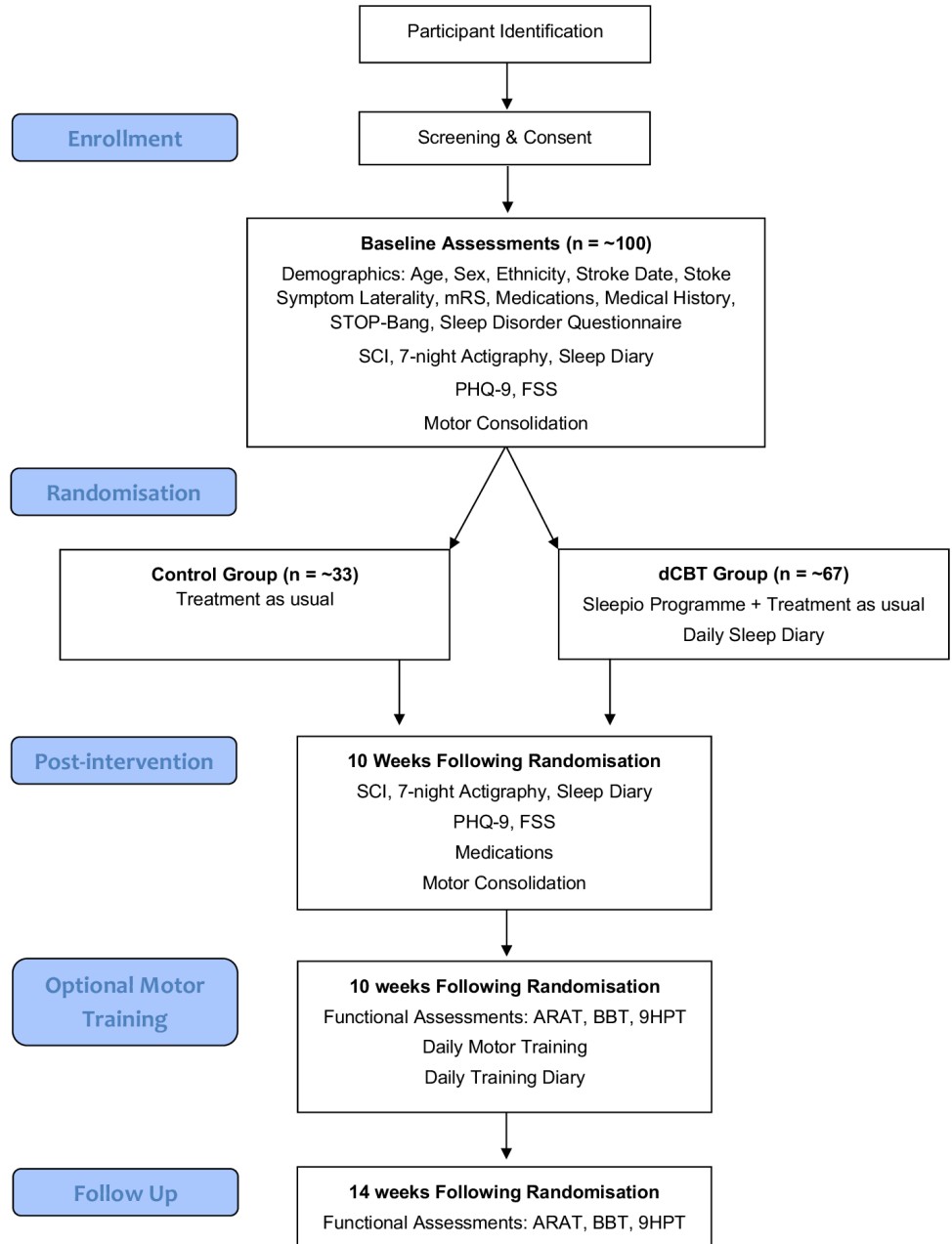

**Figure 1** Flowchart of the trial, depicting patient recruitment, screening, intervention, assessments and optional motor training. ARAT, Action Research Arm Test; BBT, Box and Blocks Test; dCBT, Digital Cognitive behavioural Therapy; FSS, Fatigue Severity Scale; 9HPT, Nine-Hole Peg Test; mRS, Modified Rankin Scale; PHQ-9, Patient Health Questionnaire; SCI, Sleep Condition Indicator; STOP-Bang, Obstructive Sleep Apnoea Questionnaire.

## Trial design

This study will be conducted as a randomised controlled trial with two parallel study arms: (1) digital cognitive behavioural therapy for insomnia (dCBT; experimental condition) and (2) treatment as usual (treatment as usual; control condition), see figure 1. The trial will, therefore, be dCBT+treatment as usual versus treatment as usual only and is reported in accordance with the Standard Protocol Items: Recommendations for Interventional Trials checklist[37] (see online supplemental file 1). Participants will receive a version of the participant information sheet and can ask the researchers any questions

they may have. Electronic informed consent is obtained from the participant (using Jisc) before any study-specific procedures are performed (online supplemental file 2).

Baseline measures of sleep will be obtained via completion of the Sleep Condition Indicator (SCI[38]) and 7-night actigraphy recordings (measuring estimated total sleep time, wakefulness after sleep onset (WASO) and sleep fragmentation index). For the actigraphy, a waterproof monitor (MotionWatch V.8, CamNTech) will be delivered to participants to be worn on their least affected wrist. Participants will also complete a sleep diary (paper or digital), detailing approximately what time they attempted

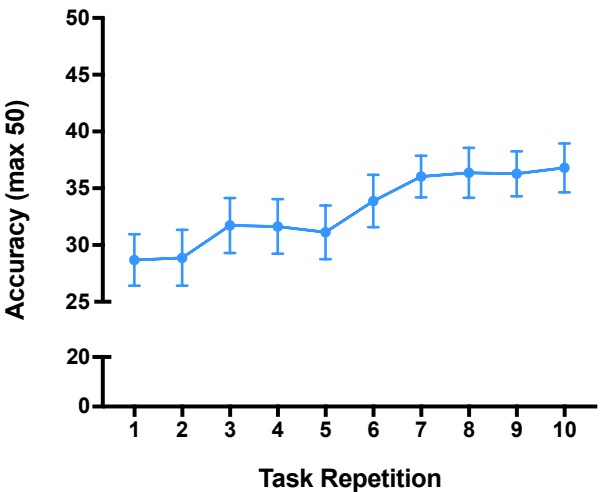

**Figure 2** Number of targets hit (maximum 50) per repetition of the GripAble motor learning task in a sample of stroke survivors with upper limb impairment. Mean scores indicated by filled circles±standard error (sample size: n=25).

sleep and woke up for each of the seven nights. Mood and fatigue will be assessed using the Patient Health Questionnaire (PHQ-9[39]) and the Fatigue Severity Scale (FSS[40 41]), respectively.

We will assess consolidation of motor learning using the commercial rehabilitation device; GripAble (GripAble Ltd). At home, participants will complete a learning task called Balloon Buddies[42] at specific times of day. The task will require participants to squeeze and release the GripAble device (a hand grip dynamometer) with their affected hand in order to direct an avatar, which appears on an associated tablet, towards targets set along a predetermined path. In each attempt of the task, participants can score up to 50 points. The score participants achieve will be displayed on the screen at the end of each task attempt. We will randomly assign participants to complete one of three different 'levels' of the task, which only differ in regards to the target path but not in difficulty. Before starting the task, the GripAble will be calibrated to each participants' individual grip and release strength/ability. Participants will be instructed to first complete the task in the evening (between 20:00 and 22:00 or at least 30 min prior to sleep). They will complete the task 10 times (recording how many targets they hit each time), lasting no longer than 30 min. Pilot data in a previous sample of stroke survivors with upper limb impairment indicate significant improvements in accuracy over 10 repetitions (figure 2). They will then be asked to perform a retest the next morning (between 8:00 and 10:00 or at least 30 min after wake), completing the same task a further five times. Again, we will ask participants to record their scores and the retest should take less than 15 min.

Age, sex, ethnicity, date of stroke/s, stroke symptom laterality, modified Rankin Scale (mRS) and current medications will also be collected. Additionally, we will use the STOP-Bang questionnaire[43] to assess the potential risk of obstructive sleep apnoea and a further sleep disorder questionnaire[44] to screen for sleep conditions other than insomnia. All demographic information will be collected and stored using online questionnaires (Jisc) or over the phone depending on the participants' preference. Additionally, all equipment will either be sent to the participants' homes or delivered by a researcher (depending on location and preference) and will be collected in person on completion or return postage will be organised.

### Randomisation and blinding

Following baseline assessments, participants will be randomised at a 2:1 ratio (experimental:control) using an online randomisation software (rando.la), with minimisation of between group differences in age, sex, baseline SCI score and time since stroke (months). Once randomised, each participant will be assigned a sequentially numbered study ID. It is not possible to blind the researchers involved in the day-to-day running of the study or the participants to their intervention group. Analysis will be carried out by an individual not involved in day-to-day research activity and not explicitly informed of the experimental condition.

### Intervention

For the intervention group, participants will be asked to continue any of their usual treatments. In addition to this, they will be provided with access to the commercially available dCBT programme, Sleepio (Big Health Ltd). The programme involves six sessions of automated dCBT, with a virtual professor, and access to a daily online sleep diary. dCBT sessions become available 7 days following completion of the previous session. Participants may also interact with an online community and library of resources. Participants will be provided with a web link to the programme and will need to sign up by entering their name, email address and a predetermined participant ID. As part of the dCBT programme, participants will be required to complete a daily online sleep diary for the duration of the programme and watch weekly videos pertaining to dCBT. Based on feedback from previous studies, they will also receive a booklet with additional details that may be useful for accessing the programme.

The control group will be asked to continue to adhere to their treatment as usual, with no further intervention measures specific to the research. The exact nature of this treatment will vary across participants. At post-intervention, we will ask whether participants in the control group received any treatment for their sleep. Participants in the control group will be provided the opportunity to access the dCBT programme after completing the study; however, no data will be recorded from this.

### Post-intervention

Approximately 10 weeks following randomisation, assessments of sleep (SCI, 7-night actigraphy, sleep diary), mood (PHQ-9), fatigue (FSS) and motor consolidation (GripAble task) will be repeated. Participants will be

randomly assigned a different level of motor learning task, so that the movement pattern is different to that which they learnt at baseline. Any participants who have not completed the dCBT programme by 10 weeks (or do not wish to complete it) will still be given the opportunity to complete the post-intervention assessments. Once completed, any equipment will be picked up by a researcher or return postage organised (dependent on participant preference, location and desire to complete voluntary training, see next section).

## Optional motor training

Before completion of the post-intervention assessments, all participants—regardless of group allocation—will be offered 4 weeks of home-based upper limb training. This training will involve daily use of the GripAble rehabilitation device and target a wide range of hand and arm movements. Participants will be asked to attempt to perform 1 hour of training per day, either in one session or broken up into shorter sessions throughout the day. Depending on their level of impairment, participants will be instructed by a researcher, which games on the device will be most suitable and how to access them, but participants are free to explore the device as they wish. We will also ask participants to complete a training diary to record how many times they use the device each day and to estimate how long they spent practising each day in total. Participants access the device through a research account, which allows the research team access to a dashboard of data. We will use the dashboard information to record the amount of time spent on each task and the number of movement repetitions performed each day. If participants decide to engage in this training, assessments of upper limb function will be conducted using the Action Research Arm Test (ARAT[45]), Box and Blocks Test (BBT[46]), The Nine-Hole Peg Test (9HPT[47]) and range of movement using the GripAble device (pronation/supination, wrist flexion/extension, radial/ulnar deviation and grip squeeze (strength) and release (%)). These assessments will be collected in-person, during the post-intervention visit when actigraphy and GripAble devices are delivered and repeated approximately 4 weeks later after completion of the training programme. Data from the optional training intervention will help to inform protocols and effect size estimates of future studies combining sleep improvement and at-home rehabilitation programmes.

## Outcome measures

The primary outcome of this study will be self-reported insomnia symptoms pre to post intervention compared with treatment as usual, measured via SCI scores.[38] SCI data will be collected during baseline assessments and approximately 10 weeks after randomisation, following either the dCBT intervention or treatment as usual. SCI has been shown to be a robust screening tool for the clinical evaluation of insomnia, with an estimated magnitude change of 7 scale points representing a meaningful improvement in insomnia symptoms.[48] Higher values on

the SCI indicate fewer symptoms of insomnia and a score of 16 or below suggests probable insomnia.[38]

Secondary outcome measures include: change in overnight motor memory consolidation between groups; changes in actigraphy-derived sleep measures (estimated total sleep time, WASO and sleep fragmentation index) between groups; correlations between changes in SCI and actigraphy-derived sleep measures and overnight motor memory consolidation in the dCBT group and changes in symptoms of depression and fatigue between the dCBT and control groups. Overnight motor consolidation will be determined by changes in performance on the GripAble motor learning task from evening training to morning retest,[49] measured at baseline and approximately 10 weeks following randomisation. Actigraphy-based sleep measures at baseline and approximately 10 weeks following randomisation will be extracted and analysed using custom software, MotionWare (CamN-Tech Ltd) and corroborated with sleep diary entries. The PHQ-9 and FSS, measured at baseline and approximately 10 weeks following randomisation, will be used to measure changes in depression and fatigue between the dCBT and control groups. In addition to these secondary outcomes, we intend to measure the adherence and efficacy of at-home motor training during the optional intervention as an exploratory outcome. Training diaries and changes in upper limb function (ARAT, BBT and 9HPT scores) will help to determine feasibility and potential effect size estimates.

## Data analysis
### Primary outcome

The primary outcome, SCI score at post-intervention, will be analysed using intention-to-treat (ITT). An analysis of covariance (ANCOVA) will be used to test whether self-reported symptoms of insomnia are improved for the dCBT group in comparison to the control group. The model will test for differences in SCI scores between groups at the post-intervention time point (~10 weeks following randomisation), with baseline score, sex and time since stroke added as covariates. All ANCOVAs will be run in general linear model format.

### Secondary/exploratory outcomes

Secondary outcomes will be tested using a complete-case ITT population, including all participants for whom data are available at the post-intervention timepoint (irrespective of whether they completed the intervention or not). To test for changes in motor consolidation, we will determine the difference in performance from the end of evening training relative to the morning retest on the GripAble motor learning task at each time point. To determine whether motor consolidation improves for the dCBT group compared with control, we will use an ANCOVA. The model will test for between-group differences at post-intervention (~10 weeks following randomisation), with baseline consolidation score as a covariate. Changes in actigraphy-derived sleep measures will also

be analysed using an ANCOVA, testing for differences between groups at 10 weeks following randomisation with baseline scores added as covariates. To explore whether the magnitude of changes in sleep measures relate to changes in motor consolidation, we will use correlational analysis (either Spearman's or Pearson's correlations as appropriate), testing for associations between changes in SCI and actigraphy-derived sleep measures (estimated total sleep time, WASO and sleep fragmentation index) with changes in motor consolidation. Changes in symptoms of depression and fatigue in the dCBT group compared with the control group will also be analysed using an ANCOVA to examine differences between groups on the PHQ-9 and FSS at follow-up, with baseline score added as a covariate. For all analyses, we will include covariates such as age, sex, time since stroke, etc as required.

### Patient and public involvement

A patient and public involvement (PPI) contributor (RF) has been involved in the setup of the study, including providing feedback on the dCBT programme and assisting in interviewing/appointing the research assistant who will be responsible for day-to-day management of the study. They have critically reviewed the protocol and will be included as a co-author in all published material from this study.

Throughout our previous studies of sleep after stroke and brain injury, we have discussed sleep concerns with patients in hospital and in the community (eg, at stroke user group meetings and stroke awareness day events). We have encouraged patients to try out and provide feedback on the motor learning task at various public/patient engagement events before and throughout the design of the study. We used this feedback to design the instructions we give to participants and to decide on which level of difficulty would be appropriate. Stroke survivors have also been involved in reviewing participant facing documents but will not be directly involved in recruitment of study participants or conduct of the study. We will seek further assistance from PPI contributors to help with preparation of lay summaries and dissemination of our results to ensure that these are meaningfully communicated to the stroke community.

### Study monitoring

Regular monitoring of participants will be performed by the research team as appropriate. Data will be evaluated for compliance with the protocol and accuracy in relation to source documents. The study may also be monitored or audited by responsible individuals from the University Sponsor and the NHS Trust(s).

At 10 weeks following randomisation, we will check with participants for the potential occurrence of any adverse effects. Any adverse events (AEs) that occur during enrolment in the trial will be documented and participants will be advised on the best course of action. This may include withdrawing from certain assessments or contacting their general practitioner. Expected minor AEs include: mild skin irritation due to wearing the actigraphy monitor, tiredness/worsening of sleep as part of the sleep restriction included in the dCBT programme, minor distress caused by completing sleep and mood questionnaires, etc.

There are no anticipated serious AEs (SAEs) associated with this trial. Any SAEs that occur will be reported to the appropriate research ethics committee, where in the opinion of the chief investigator the event was 'related' (resulted from administration of any of the research procedures) and 'unexpected' in relation to those procedures. Reports of related and unexpected SAEs will be submitted within 15 working days of the chief investigator becoming aware of the event, using the Health Research Authority (HRA) report of SAE form (see HRA website).

### Data management

Access will be granted to authorised representatives from the sponsor and host institution for monitoring and/or audit of the study. The participant will be referred to by their study ID, not by name, on all documents apart from the consent form. All trial data will be entered on paper and/or an electronic document (eg, Microsoft Excel) and stored safely in confidential conditions.

### Early discontinuation/withdrawal

During the trial, a participant may choose to withdraw early at any time. In addition, an investigator may discontinue a participant from the trial if considered necessary for any reason including, but not limited to: ineligibility (either arising during the study or retrospectively having been overlooked at screening), significant protocol deviation, significant non-compliance with study requirements, medical instability and subsequent stroke affecting motor function or requiring readmission. The type of withdrawal and reason for withdrawal will be recorded. If permitted and appropriate, data obtained up until the point of withdrawal to be retained for use in the study analysis. No further data or samples would be collected after withdrawal. Withdrawn participants will be replaced providing there is sufficient time to conduct all follow-up assessments within the study period.

### Ethics and dissemination

This study has received both Health Research Authority (HRA) and Health and Care Research Wales (HCRW) approval (IRAS ID: 306291). The study has also received approval from the National Research Ethics Service (22/EM/0080). The trial has been registered on a publicly accessible database prior to the recruitment of the first participant (clinicaltrials.gov). Results will be uploaded to clinicaltrials.gov within 12 months of the end of trial declaration by the CI or their delegate and disseminated via presentations at scientific conferences, peer-reviewed publication, public engagement events, stakeholder organisations and other forms of media where appropriate. The investigators will be involved in reviewing

drafts of the manuscripts, abstracts, press releases and any other publications arising from the study.

**Author affiliations**

[1]Wellcome Centre For Integrative Neuroimaging, FMRIB, Nuffield Department of Clinical Neurosciences, University of Oxford, Oxford, UK

[2]Patient and Public Involvement (PPI) Author, Oxford, UK

[3]Big Health Ltd, London, UK

[4]Sir Jules Thorn Sleep & Circadian Neuroscience Institute, Nuffield Department of Clinical Neurosciences, University of Oxford, Oxford, UK

[5]Oxfordshire Stroke Rehabilitation Unit (OSRU), Oxford Health NHS Foundation Trust, Oxford, UK

[6]Oxford Centre for Enablement (OCE), Oxford University Hospitals NHS Foundation Trust, Oxford, UK

[7]MRC Stroke Unit, Oxford Centre for Enablement (OCE), Oxford University Hospitals NHS Foundation Trust, Oxford, UK

[8]Department of Rehabilitation and Therapy Services, National Hospital for Neurology and Neurosurgery, University College London NHS Foundation Trust, London, UK

[9]Wellcome Centre for Human Neuroimaging, Queen Square Institute of Neurology, University College London (UCL), London, UK

[10]Wolfson Centre for the Prevention of Stroke and Dementia, Nuffield Department of Clinical Neurosciences, University of Oxford, Oxford, UK

[11]Cognitive Neuropsychology Centre, Department of Experimental Psychology, Oxford University, Oxford, UK

**Acknowledgements** We would like to thank the NIHR CRN Stroke for their support in setting up, recruiting for, and facilitating the trial.

**Contributors** MKF conceived and designed the protocol with input from MW, BR, RF, CAE, SDK, ALH, BS and HJ-B. EG, AP, RT, SA and ND facilitate recruitment and data collection. MW and BR are involved in the day-to-day running of the trial. MW drafted the manuscript. All authors edited, revised and approved the final version of the manuscript.

**Funding** This study is funded by Guarantors of Brain (Fellowship to MKF) and the Wellcome Trust (Principal Research Fellowship to HJB; 222446/Z/21/Z) and supported by the NIHR Oxford (Oxford University Hospitals) and Oxford Health Biomedical Research Centres (BRC). These funding sources had no role in the design of this study and will not have any role during its execution, analyses, or interpretation of data.

**Competing interests** CAE is co-founder and Chief Scientist of Big Health Ltd, is salaried and is a shareholder. ALH is employed by Big Health Ltd, is salaried and a shareholder. Big Health Ltd is supplying Sleepio free of charge to the study.

**Patient and public involvement** Patients and/or the public were involved in the design, or conduct, or reporting, or dissemination plans of this research. Refer to the Methods section for further details.

**Patient consent for publication** Consent obtained directly from patient(s).

**Provenance and peer review** Not commissioned; externally peer reviewed.

**Data availability statement** De-identified data will available upon reasonable request upon the completion of the trial.

**ORCID iDs**

Matthew Weightman http://orcid.org/0000-0003-4379-2725

Barbara Robinson http://orcid.org/0000-0002-1721-7682

Simon D Kyle http://orcid.org/0000-0002-9581-5311

Melanie K Fleming http://orcid.org/0000-0003-2232-9598

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
