## [Reviewer comments · BMJ Open]

ARTICLE DETAILS

TITLE (PROVISIONAL)	Improving sleep and learning in rehabilitation after stroke, part 2 (INSPIRES2): study protocol for a home-based randomised control trial of digital cognitive behavioural therapy (dCBT) for insomnia
AUTHORS	Weightman, Matthew; Robinson, Barbara; Fallows, Ricky; Henry, Alasdair; Kyle, Simon; Garratt, Emma; Pick, Anton; Teal, Rachel; Ajina, Sara; Demeyere, Nele; Espie, Colin; Seymour, Ben; Johansen-Berg, Heidi; Fleming, Melanie K.

VERSION 1 – REVIEW

REVIEWER	Korostovtseva, Lyudmila Almazov National Medical Research Centre, Hypertension Department, Somnology Group
REVIEW RETURNED	01-Feb-2023

GENERAL COMMENTS	The paper presents a protocol of a randomized controlled clinical trial aimed at the improvement post-stroke rehabilitation by applying digital (on-line) cognitive behavioural therapy for insomnia. The background and aims of the study are clearly presented. Study design and chosen methods are appropriate. Planned sample size is substantiated. There are several considerations which require clarification: - The researchers will not be blinded, however, the authors could plan a blinded assessment of the outcome measures. This is not clearly stated.- The authors state that "Participants may be identified at any time following their stroke but will not be enrolled in the study until they have been discharged from inpatient care." Does it mean that acute, subacute and chronic stage post-stroke patients can be enrolled in the study? If any limited post-stroke period is anticipated for recruitment it should be clarified.- Will the authors apply any criteria based on routine clinical scales for stroke severity and post-stroke disability (NIHSS, mRS etc.)?- How will the authors consider any other treatment for insomnia (e.g. sleeping pills), will they be considered an exclusion criteria or will the authors adjust for it during analysis? In general, the study aims at answering important questions in post-stroke rehabilitation and is relevant. After minor revision the protocol can be published. - It is not clear how the authors will proceed with those patients who decline the intervention after randomization. In the section "Post-intervention" the authors mention "Any participants who have not completed the dCBT programme by 10 weeks (or do not wish to complete it)..." If the patient does not wish
---

	to complete it is he/she subject to withdrawal from the study? (see section "early discontinuation/withdrawal)  - How the analysis will be performed (as intention-to-treat analysis or not?) - Are there any anticipated (non-serious) adverse events specifically related to the intervention? If yes, this should be mentioned. - It would be reasonable to mention the date of the registration of clinicaltrials.gov
--	---

REVIEWER	Morris, Meg La Trobe Univ, Room 423 HS3
REVIEW RETURNED	08-Feb-2023

GENERAL COMMENTS	This is a scholarly manuscript on an important and novel topic. It is clearly written. The methods for this protocol are presented in a systematic manner using validated reporting tools. There are only a few areas needing minor revision: (1) In the introduction and throughout the authors refer to "online" and "offline" - does this mean digital healthcare (eg online using ZOOM) or is it a psychology term? Most health professionals will understand these terms in relation to digital healthcare delivery. Please can you revise the manuscript to clarify. (2) In the abstract and introduction, the term "motor learning" is referred to yet it is unclear how motor skill learning and sleep rehabilitation are related - please can you explain this more clearly. (3) The sample size is the major issue. This sample size of 100 seems to be arbitrary and the justification why only 50 per group is really inadequate - this seems to be a really small number - most stroke trials of this nature need hundreds in each group to show a difference in these sorts of outcome variables - please have a statistician revise and justify in much greater depth the sample sizes. (4) The Discussion repeats a lot of the earlier text again. Please remove, for example, the first paragraph of the Discussion and update it to include some more recent refs to other research globally on rehabilitation post stroke.
--

VERSION 1 – AUTHOR RESPONSE

REVIEWER #1

- 1) The researchers will not be blinded, however, the authors could plan a blinded assessment of the outcome measures. This is not clearly stated.

Thank you for raising this point. We did consider this, however, as randomisation is 2:1 (intervention:control), it would therefore be difficult to blind the assessment/analysis of outcome measures. However, the researchers conducting the statistical analyses have no interaction with the participants during the study.

- 2) The authors state that "Participants may be identified at any time following their stroke but will not be enrolled in the study until they have been discharged from inpatient care." Does it mean that

acute, subacute and chronic stage post-stroke patients can be enrolled in the study? If any limited post-stroke period is anticipated for recruitment it should be clarified.

Yes, we will recruit stroke survivors at all stages post-stroke, as long as they have been discharged from inpatient care and fit the rest of the inclusion/exclusion criteria. There is no specific post-stroke period restriction for recruitment in this trial (we have now made this more explicit in the participant recruitment section).

“There is no specific post-stroke period restriction for recruitment to this trial.”

Time since stroke is included as a variable in the minimisation process of the randomisation and so we hope that the two groups will not differ substantially in this variable. Regardless, time since stroke will be included as a covariate in the analysis. We have updated the data analysis section to make clear that we will include baseline factors such as this as covariates.

“The model will test for differences in SCI scores between groups at the post-intervention time point (~10 weeks following randomisation), with baseline score, sex and time since stroke added as covariates. All ANCOVAs will be run in general linear model format.”

3) Will the authors apply any criteria based on routine clinical scales for stroke severity and post-stroke disability (NIHSS, mRS etc.)?

Yes, we will collect mRS data from each participant at baseline, which is used for cohort demographic purposes. However, this measure is not incorporated into inclusion/exclusion criteria. This information is included in the ‘Study Design’ section and Figure 1.

4) How will the authors consider any other treatment for insomnia (e.g. sleeping pills), will they be considered an exclusion criteria or will the authors adjust for it during analysis?

At baseline, data regarding current medication(s) will be collected and therefore we can determine if any participants are receiving pharmacological treatments for sleep. However, this is not an exclusion criterion. Additionally, at follow up, we will ask all participants if they have received any other treatment for sleep during the trial. If possible, we will adjust for this in any analysis.

If participants have received psychological treatment for insomnia/sleep problems within the past 12 months, they will be excluded from the study. Participants will also be excluded if they have a diagnosed (untreated) sleep disorder.

5) It is not clear how the authors will proceed with those patients who decline the intervention after randomization.

Thank you for highlighting this missing information. We will conduct an intention-to-treat analysis. This has now been added to the manuscript in the Data analysis section. We will also follow the procedures outlined in the ‘Early Discontinuation/Withdrawal’ with respect to any data already collected.

“The primary outcome, SCI score at post-intervention, will be analysed using intention-to-treat (ITT).”

We will try to mitigate this problem by asking if participants are interested in a sleep improvement programme prior to consent. Therefore, any individuals who are explicitly not interested will not be enrolled in the study.

- 6) In the section "Post-intervention" the authors mention "Any participants who have not completed the dCBT programme by 10 weeks (or do not wish to complete it)..." If the patient does not wish to complete it is he/she subject to withdrawal from the study? (see section "early discontinuation/withdrawal")

As indicated in the 'Post Intervention' section, any participants who have not completed the dCBT programme will still be given the opportunity to complete the follow-up assessments and optional motor training. We are also able to gather data from Sleepio to determine participant engagement with the dCBT programme.

- 7) How the analysis will be performed (as intention-to-treat analysis or not?)

Thank you for highlighting this missing information. The primary outcome will be analysed using intention-to-treat. Secondary and exploratory outcomes will utilise a complete case intention-to-treat population, restricted to randomised participants for whom data is available at the following timepoint (but irrespective of how much they engaged in the intervention).

"The primary outcome, SCI score at post-intervention, will be analysed using intention-to-treat (ITT)."

"Secondary outcomes will be tested using a complete case ITT population, including all participants for whom data is available at the post-intervention timepoint (irrespective of whether they completed the intervention or not)."

- 8) Are there any anticipated (non-serious) adverse events specifically related to the intervention? If yes, this should be mentioned.

Yes, examples of minor AEs are mentioned in the 'Study Monitoring' section of the original manuscript. We have altered the wording slightly to make this clearer:

"Any adverse events (AEs) that occur during enrolment in the trial will be documented and participants will be advised on the best course of action. This may include withdrawing from certain assessments or contacting their General Practitioner (GP). Expected minor AEs include: mild skin irritation due to wearing the actigraphy monitor, tiredness/worsening of sleep as part of the sleep restriction included in the dCBT programme, minor distress caused by completing sleep and mood questionnaires etc."

- 9) It would be reasonable to mention the date of the registration of clinicaltrials.gov

This information is provided in the abstract as per BMJ Open preferred formatting. We have now also added this in the participant recruitment section.

"The study was registered on 22nd August 2022 (prior to recruitment of the first participant; NCT05511285)"

REVIEWER #2

- 1) In the introduction and throughout the authors refer to "online" and "offline" - does this mean digital healthcare (eg online using ZOOM) or is it a psychology term? Most health professionals will understand these terms in relation to digital healthcare delivery. Please can you revise the manuscript to clarify.

In the introduction (paragraph 2) we refer to both online and offline improvements in motor learning. These are common neuroscience terms, denoting whether changes happen during learning itself or occur during rest. However, to avoid any confusion with the digital healthcare aspects of the trial we have reworded the section, see below:

“Improvements in motor performance not only occur during practise itself, but also during periods of rest, including sleep (13). In recent years, a large body of evidence has emerged indicating these latter improvements in performance, better known as consolidation, are sleep-dependent (14-21).”

Any other mentions of ‘Online’ are followed by a descriptor making it clear that it is referring to a computer programme or digital tool e.g., online questionnaires, online randomisation, online sleep diary, online community. There are no further mentions of ‘Offline’ in the manuscript. We hope this clarifies your concern.

- 2) In the abstract and introduction, the term "motor learning" is referred to yet it is unclear how motor skill learning and sleep rehabilitation are related - please can you explain this more clearly.

Thank you for this comment. The link between sleep and consolidation of motor learning is the main rationale for this study and therefore it is paramount that this information is clear.

In short, we suggest (as many other have previously) that the consolidation of motor learning is sleep-dependent. We also present evidence to suggest that stroke survivors typically sleep less well. Given these points, it stands to reason that disrupted sleep post-stroke may impair the consolidation of motor learning (which is a fundamental component of motor rehabilitation post stroke) between therapy sessions and ultimately impacting on their functional recovery. Our trial protocol is therefore hoping to discover if dCBT for insomnia can improve sleep after stroke and whether this leads to an improvement in the consolidation of motor learning.

We have integrated some of the points that were initially in the ‘Discussion’ section into the Introduction to help link these two components more clearly.

“Stroke is one of the leading causes of long-term disability worldwide, often resulting in prolonged cognitive, motor, and emotional problems (1). Many stroke survivors also experience long-term sleep disruption (2-4), with the prevalence of post-stroke insomnia, a sleep disorder characterised by difficulties initiating and maintaining sleep, estimated to be as high as 60% (5-7). Poor sleep after stroke is frequently associated with motor/cognitive impairments, mood disorders such as anxiety and depression, and a reduced quality of life (8-10). There is also appreciable evidence to suggest that those who have poorer sleep show worse rehabilitation outcomes (8,11), which we hypothesise may be, at least in part, due to the importance of sleep for learning and memory processes.

Motor learning – the process of learning new, or relearning old, motor skills – is an integral part of post-stroke rehabilitation (12). Improvements in motor performance not only occur during practise itself, but also during periods of rest, including sleep (13). In recent years, a large body of evidence has emerged indicating these latter improvements in performance, better known as consolidation, are sleep-dependent (14-21).”

- 3) The sample size is the major issue. This sample size of 100 seems to be arbitrary and the justification why only 50 per group is really inadequate - this seems to be a really small number - most stroke trials of this nature need hundreds in each group to show a difference in these sorts of outcome variables - please have a statistician revise and justify in much greater depth the sample sizes.

This sample size was calculated based on prior work in our lab (Fleming et al., 2023) involving dCBT in stroke survivors. To summarise, in 84 community dwelling stroke survivors (n = 48 dCBT, n = 36 control) we found significant improvements in self-reported symptoms of insomnia (SCI) and mood following CBT in comparison with provision of sleep hygiene information, with a medium effect size (SCI $p \leq 0.02$, $\eta^2 = 0.07 - 0.12$, pooled mean difference = -3.35). We also used other studies involving dCBT in people who suffer from insomnia to determine our sample size, which is described in the 'Sample Size Distribution' section of the manuscript.

We estimated a sample size of 76 would be required to achieve the desired effect size $d = 0.7$). The final sample size of 100 was selected to accommodate any drop out from the study based on our previous experience with the same CBT programme). Additionally, group allocation will occur on a 2:1 (intervention:control) basis so group sizes will be approximately 66:33.

Hopefully this information is sufficient.

- 4) The Discussion repeats a lot of the earlier text again. Please remove, for example, the first paragraph of the Discussion and update it to include some more recent refs to other research globally on rehabilitation post stroke.

Thank you, this has now been removed as per the editor/s request. Some elements have now been integrated into the introduction to improve the clarity of our rationale.

VERSION 2 – REVIEW

REVIEWER	Korostovtseva, Lyudmila Almazov National Medical Research Centre, Hypertension Department, Somnology Group
REVIEW RETURNED	22-Mar-2023
GENERAL COMMENTS	Thank you for clear answers. The comments are properly addressed.
REVIEWER	Morris, Meg La Trobe Univ, Room 423 HS3
REVIEW RETURNED	20-Feb-2023
GENERAL COMMENTS	Thank you for the revisions - all of my questions have been addressed.